

# Identification of stem rust resistance genes in wheat cultivars in China using molecular markers

Xiaofeng Xu[1],[*], Depeng Yuan[1],[*], Dandan Li[1], Yue Gao[1], Ziyuan Wang[1], Yang Liu[1], Siting Wang[1], Yuanhu Xuan[1], Hui Zhao[2], Tianya Li[1] and Yuanhua Wu[1]

[1] College of Plant Protection, Shenyang Agricultural University, Shenyang, China
[2] Henan Academy of Agricultural Science, Institute of Plant Protection, Henan, China
[*] These authors contributed equally to this work.

## ABSTRACT

Wheat stem rust caused by *Puccinia graminis* f. sp. *tritici* Eriks. & E. Henn. (*Pgt*), is a major disease that has been effectively controlled using resistance genes. The appearance and spread of *Pgt* races such as Ug99, TKTTF, and TTTTF, which are virulent to most stem rust-resistant genes currently deployed in wheat breeding programs, renewed the interest in breeding cultivars resistant to wheat stem rust. It is therefore important to investigate the levels of resistance or vulnerability of wheat cultivars to *Pgt* races. Resistance to *Pgt* races 21C3CTHQM, 34MKGQM, and 34C3RTGQM was evaluated in 136 Chinese wheat cultivars at the seedling stage. A total of 124 cultivars (91.2%) were resistant to the three races. Resistance genes *Sr2*, *Sr24*, *Sr25*, *Sr26*, *Sr31*, and *Sr38* were analyzed using molecular markers closely linked to them, and 63 of the 136 wheat cultivars carried at least one of these genes: 21, 25, and 28 wheat cultivars likely carried *Sr2*, *Sr31*, and *Sr38*, respectively. Cultivars "Kehan 3" and "Jimai 22" likely carried *Sr25*. None of the cultivars carried *Sr24* or *Sr26*. These cultivars with known stem rust resistance genes provide valuable genetic material for breeding resistant wheat cultivars.

# INTRODUCTION

Wheat stem rust caused by *Puccinia graminis* Per. f. sp. *tritici* Eriks. & E. Henn. (*Pgt*) is a devastating disease that has caused severe yield losses worldwide. Since the deployment of stem rust-resistant wheat cultivars in the second half of the 20th century, stem rust has been successfully controlled in most wheat cultivating areas (*Chen et al., 2015*). However, a new race of the stem rust pathogen (Ug99), identified in Uganda in 1999 and highly virulent to resistance gene *Sr31*, was designated as TTKSK under the North American nomenclature system (*Pretorius et al., 2000*). Within a few years, virulence of TTKSK to other important stem rust resistance genes (e.g., *Sr24*, *Sr36*, *Sr9h*, *Sr31 + Sr24*, *Sr31 + Sr36*, and *Sr31 + SrTmp*) was detected (*BGRI, 2017*; *Jin et al., 2008*, *2009*; *Pretorius et al., 2012*; *Rouse et al., 2014*), and 13 variants of Ug99 have now been documented across wheat growing regions in 13 countries (*Food and Agriculture Organization of the United Nations (FAO), 2017*).

Corresponding authors
Tianya Li, Litianya11@syau.edu.cn
Yuanhua Wu, wuyh7799@163.com

Realizing the disastrous threat on world food security posed by the Ug99 race group, Nobel Peace Prize laureate Norman Borlaug called for a coordinated global campaign to reduce wheat rust epidemics and mitigate the potential impact on food security. The resistance of worldwide wheat accessions (over 200,000) to the Ug99 group was screened in Kenya (*He, Xia & Chen, 2008*). The results indicated that only 5–15% of the wheat accessions grown globally were resistant to Ug99, and only two of the 118 Chinese wheat cultivars ("Jimai 20" and "Linmai 6") were resistant to Ug99. The high susceptibility (85–95%) of wheat lines to Ug99 highlighted the potential threat of this group to wheat production worldwide. Furthermore, other broadly virulent *Pgt* races caused wheat stem rust epidemics in recent years. The new race TKTTF (from a genetic lineage distinct from that of Ug99) virulent to the widely grown wheat cultivar "Digalu," caused yield losses close to 100% in Southern Ethiopia during 2013–2014 (*Olivera et al., 2015*). In 2016, a new and unusually devastating strain of *Pgt* named TTTTF (virulent to *Sr9e* and *Sr13*) caused the largest outbreak and epidemics of wheat stem rust in Sicily since the 1950s (*Bhattacharya, 2017*), as tens of thousands of hectares of both durum wheat and bread wheat were infected. Thus, wheat stem rust seems to have returned.

The most effective way to control wheat stem rust is by using resistant genes against this disease to breed and propagate resistant varieties (*Pathan & Park, 2007*). However, an important issue in the use of resistant varieties is that the simplification of the resistance source may be overcome by variation in the pathogen, resulting in the loss of resistance. Understanding resistance gene content of wheat varieties can effectively avoid this situation, and provide a basis for the reasonable distribution of varieties. Moreover, it is also helpful to discover new genes, enriching the gene pool, and for breeding resistant varieties. Nevertheless, the spread of new *Pgt* races and their variants threatens the safety of wheat production in China (*Li et al., 2016*). If the conditions are suitable, there is the possibility of wheat stem rust becoming a significant disease. Because the resistance of Chinese wheat varieties to the new races Ug99, TKTTF, and TTTTF is very poor, if these races spread into China they will cause massive losses in wheat production (*Cao et al., 2007*). We should therefore make full use of wheat cultivar resources to screen for resistant materials. Given the importance of understanding disease resistance genes, those against Ug99, TKTTF, and TTTTF races have been screened and identified worldwide since these races were reported.

In our previous study, the prevalence of *Sr2*, *Sr24*, *Sr25*, *Sr26*, *Sr31*, and *Sr38* in wheat cultivars from Gansu and Yunnan Province has been finished (*Li et al., 2016*; *Xu et al., 2017*), and based on it, this study was carried out. We collected 136 wheat cultivars from two different localities presenting epidemic patterns of wheat stem rust to examine their resistance level to the predominant races of *Pgt* in China. Resistance genes *Sr2*, *Sr24*, *Sr25*, *Sr26*, *Sr31*, and *Sr38* were detected using molecular markers aiming to screen and identify cultivars that are potentially resistant to emerging races (especially to Ug99, TKTTF, and TTTTF) and map the distribution of those genes in wheat regions based on wheat cultivars' resistance level to predominant races of *Pgt*. Thus far, our team have identified and characterized these resistant genes in four wheat-producing regions of

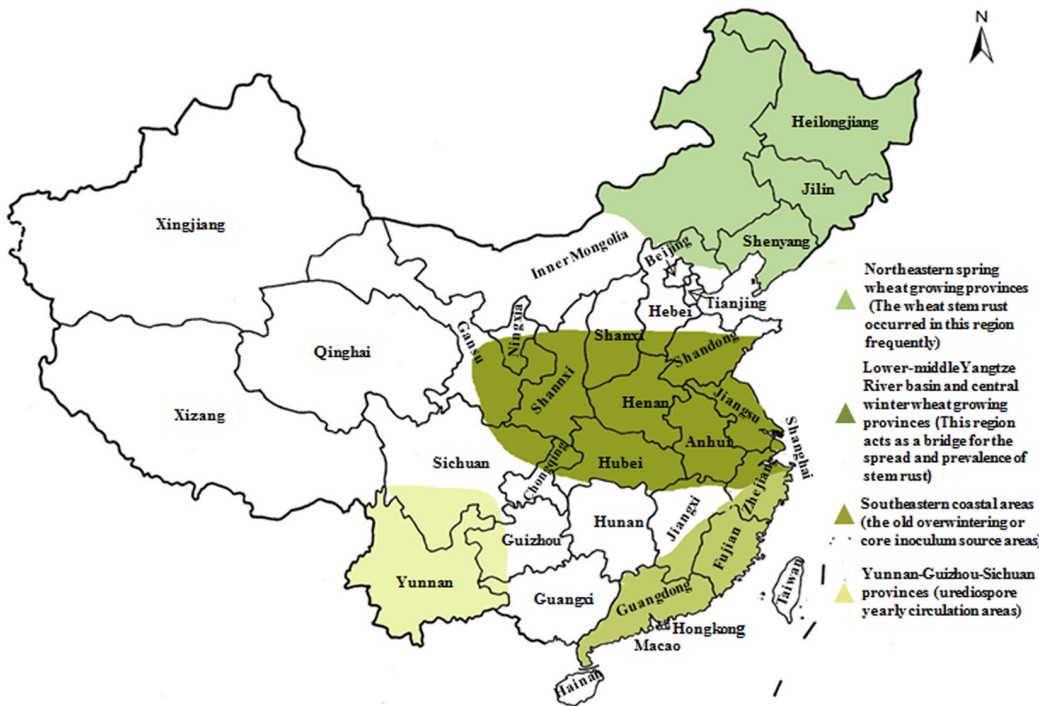

**Figure 1 Epidemic patterns of wheat stem rust in four wheat-producing regions in China.**

China (Fig. 1), which will contribute to the deployment of wheat stem rust resistance genes and control of large-scale epidemics of this disease. Additionally, this information will be important for developing potentially durable combinations of stem rust resistance genes in wheat cultivars.

## MATERIALS AND METHODS

### Wheat cultivars (lines) and *Pgt* races

A total of 136 wheat cultivars (lines) were collected from the largest wheat growing regions in China: the Northeastern spring-wheat growing provinces and the lower-middle Yangtze River basin and central winter-wheat growing provinces. All wheat accessions were provided by researchers from Heilongjiang, Inner Mongolia, Shandong, Shanxi, Anhui, Jiangsu, Beijing, and Ningxia Academies of Agricultural Sciences. A total of six monogenic wheat lines carrying individual *Sr* genes (*Sr2*, *Sr24*, *Sr25*, *Sr26*, *Sr31*, and *Sr38*), and 29 differentials for *Pgt*, including the original four Stakman differentials (Little Club, Reliance, Einkorn, and Vernal), five Chinese differentials (Mianzi 52, Huadong 6, Mini 2761, Orofen, and Rulofen), and 20 single *Sr*-gene lines from North America (*Sr5*, *Sr21*, *Sr9e*, *Sr7b*, *Sr11*, *Sr6*, *Sr8a*, *Sr9g*, *Sr36*, *Sr9b*, *Sr30*, *Sr17*, *Sr9a*, *Sr9d*, *Sr10*, *SrTmp*, *Sr24*, *Sr31*, *Sr38*, and *SrMcN*) used worldwide, were provided by the Plant Immunity Institute, Shenyang Agricultural University, China.

Races 21C3CTHQM (*Pgt* isolate Ab3), 34MKGQM (*Pgt* isolate H31), and 34C3RTGQM (*Pgt* isolate XN11) (a new race identified from the alternative host *Berberis*

sp.) were used for evaluating seedling stem rust response in the tested cultivars. These races were isolated and identified by the Plant Immunity Institute, Shenyang Agricultural University, China. The names, virulence/avirulence spectrums, and urediniospores produced method of races were described by *Li et al. (2016)* and *Xu et al. (2017)*.

## Seedling infection types

The 136 wheat accessions were planted in 10-cm diameter porcelain pots (each pot contained one cultivar represented by eight to 10 seedlings). A mixture of urediniospores and dried talc (1 g), in a ratio of 1:20 (v:v), was sprayed onto the fully expanded primary leaves of seedling (seven to eight days old) moistened with 0.05% Tween-20. The detail inoculation and cultivation methods followed *Xu et al. (2017)*. Three replicates of the seedling assays were performed for each *Pgt* race. Infection types (ITs) were assessed two weeks after inoculation using the 0–4 IT scale, as described by *Stakman, Stewart & Loegering (1962)*.

## DNA extraction

DNA was extracted from the young leaves of seven-day old seedlings grown to the one-leaf stage, using a DNA extraction kit (Sangon Biotech, Shanghai, China). PCR amplifications were followed *Xu et al. (2017)*. Primers were synthesized by Sangon Biotech (China) (Table 1), and PCR amplification conditions were as described in previous studies (Table 1). Fragments of the targeted genes were detected by electrophoresis using 2% (W/V) agarose gels and then gels were observed under UV light.

# RESULTS

## Wheat seedling resistance

The ITs produced by wheat cultivars to races 21C3CTHQM, 34MKGQM, and 34C3RTGQM are listed in Table 2. A total of 124 (91.2%) wheat cultivars were resistant to the three races (ITs 2, 1+, or lower) while the remaining 12 were susceptible (ITs 3-, 3, 3+, and 4) (Fig. 2). A total of 48 wheat cultivars (35.3%) showed IT 0 to all tested races (Fig. 2) and 127 showed resistance to the new race 34C3RTGQM.

## Detection of stem rust resistance genes using molecular markers

### *Sr2* screening

The adult plant resistant gene *Sr2*, which provides a durable broad-spectrum to *Pgt* is difficult to screen under field conditions (*Hayden, Kuchel & Chalmers, 2004*). However, the *Sr2*-closely linked microsatellite marker *Xgwm533*, developed by *Hayden, Kuchel & Chalmers (2004)*, typically amplifies a 120-bp fragment from wheat lines known to carry *Sr2*. In the present study, we used this marker to detect *Sr2* and 21 of the 136 wheat varieties showed the *Sr2* fragment (Fig. 1A; Table 2), suggesting that those wheat varieties carry *Sr2*.

### *Sr24* screening

Gene *Sr24* is effective against some *Pgt* races in China and it was derived from *Thinopyrum ponticum*. It is widely used in wheat breeding though it has become susceptible to some

**Table 1 PCR primers and conditions for the amplification of the tested markers.**

| Marker | Primers | PCR conditions | |
|---|---|---|---|
| | | Temperature (°C)/time | Number of cycles cycle |
| Xgwm533 | 5′-GTTGCTTTAGGGGAAAAGCC<br>5′-AAGGCGAATCAAACGGAATA | 92/3 min | One |
| | | 92/30 s; 62/30 s; 72/30 s | 1 °C Reducing/cycle for seven cycles |
| | | 92/30 s; 62/30 s; 72/30 s | 47 |
| Sr24#12 | 5′-CACCCGTGACATGCTCGTA<br>5′-AACAGGAAATGAGCAACGATGT | 94/3 min | One |
| | | 94/30 s; 65/30 s; 72/40 s | 1 °C Reducing/cycle for seven cycles |
| | | 94/30 s; 58/30 s; 72/40 s | 30 |
| | | 20/1 min | One |
| Gb | 5′-CATCCTTGGGGACCTC<br>5′-CCAGCTCGCATACATCCA | 94/3 min | One |
| | | 94/30 s; 60/30 s; 72/40 s | 30 |
| | | 20/1 min | One |
| Sr26#43 | 5′-AATCGTCCACATTGGCTTCT<br>5′-CGCAACAAAATCATGCACTA | 94/3 min | One |
| | | 94/30 s; 56/30 s; 72/40 s | 30 |
| | | 20/1 min | One |
| SCSS30.2$_{576}$ | 5′-GTCCGACAATACGAACGATT<br>5′-CCGACAATACGAACGCCTTG | 95/5 min; 60/1 min; 72/30 s | One |
| | | 95/1 min; 60/1 min; 72/30 s | 35 |
| | | 72/10 min | One |
| Iag 95 | 5′-CTCTGTGGATAGTTACTTGATCGA<br>5′-CCTAGAACATGCATGGCTGTTACA | 94/3 min | One |
| | | 94/30 s; 55/60 s; 72/70 s | 30 |
| | | 25/60 s | One |
| VENTRIUP-LN2 | 5′-AGGGGCTACTGACCAAGGCT<br>5′-TGCAGCTACAGCAGTATGTACACAAAA | 94/45 s | One |
| | | 94/45 s; 65/30 s; 72/7 min | 30 |
| | | 72/1 min | One |

Ug99 variants (*Jin et al., 2008*). *Mago et al. (2005)* reported that marker *Sr24#12*, linked to *Sr24*, was associated with the 3Ag/1BS Amigo-type translocation, and this marker can amplify a 500-bp fragment in the wheat variety "Westonia/Sr24." Using a diverse collection of wheat germplasm, *Yu et al. (2010)* showed that this 500-bp PCR fragment was amplified in wheat germplasm carrying *Sr24*. In the present study, 500-bp fragments were amplified in the wheat line "LcSr24Ag," suggesting it carries *Sr24* (Fig. 3B) but no fragment was amplified in the other tested varieties (Table 2).

### *Sr25* and *Sr26* screening

Ug99-effective-genes *Sr25* and *Sr26* were transferred into wheat from *T. ponticum*. These two genes were firstly backcrossed into Australian wheat, and some old varieties may carry these genes in China (*Cao et al., 1994*; *Knott, 1961*). We used markers *Gb* (amplifies a 130-bp fragment) and *Sr26#43* (amplifies a 207-bp fragment), which are closely linked to genes *Sr25* and *Sr26*, respectively (*Liu et al., 2010*), to screen these genes in the 136 accessions. The 130-bp fragment was only amplified in Kehan 3 and Jimai 22 (Fig. 3C; Table 2), indicating that only these two wheat varieties carry *Sr25*; the other tested wheat varieties lack *Sr25* and *Sr26* (Fig. 3D; Table 2).

**Table 2 Seedling infection types and resistance genes.**

| Cultivars | Province | Pedigree | Infection types[a] | | | Sr2 | Sr24 | Sr25 | Sr26 | Sr31 | Iag95 | Sr38 |
|---|---|---|---|---|---|---|---|---|---|---|---|---|
| | | | 21C3CTHQM | 34MKGQM | 34C3RTGQM | Xgwm533 | Sr24#12 | Gb | Sr26#43 | SCSS30.2$_{576}$ | Iag95 | VENTRIUP-LN2 |
| Xinkehan 9 | Heilongjiang | Kefeng 2/Ke74F$_3$-249-3 | ;1 | 1 | ; | –[b] | – | – | – | – | – | – |
| Kehan 2 | Heilongjiang | Jiusan 80 jian 119/Nongda75-65533 | ;1 | 1 | 1 | – | – | – | – | – | – | + |
| Kehan 3 | Heilongjiang | Ke 61F$_3$-199/Agropyron glaucum | 0 | 0 | 0 | – | – | + | – | – | – | – |
| Kehan 4 | Heilongjiang | Kezhen/Kehong | ;1 | 1 | 1+ | – | – | – | – | – | – | + |
| Kehan 8 | Heilongjiang | Ke65F$_2$-196-7/Rulofen | ; | 0 | 0 | – | – | – | – | – | – | – |
| Kehan 9 | Heilongjiang | Kefeng 2/Ke 74F$_3$-249-3 | ; | 0 | 0 | – | – | – | – | – | – | – |
| Kehan 10 | Heilongjiang | Kefeng 2//T808/Ke 69-513 | ; | 1 | 0 | – | – | – | – | – | – | – |
| Kehan 11 | Heilongjiang | Ke 73-402/Bei 74-205 | ; | 0 | 0 | + | – | – | – | – | – | – |
| Kehan 12 | Heilongjiang | Ke 68-88/Ke 68-585-13 | ; | 1 | 1 | – | – | – | – | – | – | + |
| Kehan 13 | Heilongjiang | Kefeng 3/Kehan 8 | ;1 | 0 | 0 | – | – | – | – | – | – | – |
| Kehan 14 | Heilongjiang | Ke 80-10-1/Ke 81 hou 88-0-1 | ; | 1– | 1 | – | – | – | – | – | – | + |
| Kehan 15 | Heilongjiang | Ke 86F$_2$-172/Ke 86F$_5$-325-3 | 0 | 0 | 0 | – | – | – | – | – | – | – |
| Kehan 16 | Heilongjiang | Jiusan 79F5-541/Ke 80 yuan 229//Ke 76-750/76F4-779-5//Ke76-413 | 0 | 0 | 0 | – | – | – | – | – | – | + |
| Kehan 18 | Heilongjiang | Jiusan 1989/Kefeng 5 | 0 | 0 | 0 | – | – | – | – | – | – | – |
| Kehan 19 | Heilongjiang | Ke 90-99/MY4490 | 1 | 0 | 0 | – | – | – | – | – | – | + |
| Kehan 20 | Heilongjiang | Ke 89-46/Cundo | ;1 | 0 | 0 | – | – | – | – | – | – | + |
| Kehan 21 | Heilongjiang | Ke89F$_6$ nan-2/Ke 89F$_1$-1237 | 1 | 2 | 1 | – | – | – | – | – | – | – |
| Kefeng 6 | Heilongjiang | Ke 85-869/Ke 85-784 | ;1 | ; | 0 | – | – | – | – | – | – | – |
| Kefeng 7 | Heilongjiang | Ke 84F$_5$-250-1/84F$_5$-668 | ; | ; | 0 | – | – | – | – | – | – | – |
| Kefeng 8 | Heilongjiang | Kehan 12/Ke 82-371 | 0 | 1 | ; | – | – | – | – | – | – | – |
| Longfu 1 | Heilongjiang | Xinshuguang 3/Liaochun 8 | 0 | 1 | ; | – | – | – | – | – | – | – |
| Longfu 2 | Heilongjiang | Longxi 35/Ke 250 | ;1 | 1 | ; | – | – | – | – | – | – | – |
| Longfu 3 | Heilongjiang | Longfu 77-4096/S-A-25 | 0 | 0 | 0 | – | – | – | – | – | – | – |
| Longfu 4 | Heilongjiang | Heiza 266/Ke 79F3-392 | ; | 1 | 1 | – | – | – | – | – | – | – |
| Longfu 5 | Heilongjiang | Jiusan B29-/32P | 0 | 0 | 0 | – | – | – | – | – | – | – |
| Longfu 6 | Heilongjiang | Longfu 2108/Haishu | 0 | 0 | 0 | – | – | – | – | – | – | – |
| Longfu 7 | Heilongjiang | Longfu 3/Gang 98-446 | ; | 0 | 0 | – | – | – | – | – | – | – |

| Cultivars | Province | Pedigree | Infection types[a] | | | Sr2 | Sr24 | Sr25 | Sr26 | Sr31 | Iag95 | Sr38 |
|---|---|---|---|---|---|---|---|---|---|---|---|---|
| | | | 21C3CTHQM | 34MKGQM | 34C3RTGQM | Xgwm533 | Sr24#12 | Gb | Sr26#43 | SCSS30.2$_{576}$ | | VENTRIUP-LN2 |
| Longfu 8 | Heilongjiang | K202 60Coγ 1000 Rad | 0 | 0 | 0 | – | – | – | – | – | – | – |
| Longfu 9 | Heilongjiang | Kejian 23 60Coγ180Gy | 0 | 0 | 0 | – | – | – | – | – | – | – |
| Longfu 10 | Heilongjiang | Ke 87-183 γ1.1 kRad | 0 | 0 | 0 | – | – | – | – | – | – | – |
| Longfu 11 | Heilongjiang | Longfu 81-8106 60 Coγ 1.1 kRad | 0 | 0 | 0 | – | – | – | – | – | – | + |
| Longfu 12 | Heilongjiang | Jia 5 60 Coγ | ; | ; | 0 | – | – | – | – | – | – | – |
| Longfu 13 | Heilongjiang | Unknown | 2 | 1 | 1 | – | – | – | – | – | – | – |
| Longfu 14 | Heilongjiang | F$_0$ (Ke 86F6-545/Hei 85-1584) γ1.0 Rad | 1 | 0 | 1 | – | – | – | – | – | – | – |
| Longfu 16 | Heilongjiang | Unknown | 0 | 0 | ;1 | – | – | – | – | – | – | – |
| Longfu 18 | Heilongjiang | Long 94-4083 mutagenesis | ; | 0 | ; | + | – | – | – | – | – | – |
| Longfu 19 | Heilongjiang | SP4/Longmai 26 | 0 | 0 | 0 | – | – | – | – | – | – | – |
| Longfu 20 | Heilongjiang | Xiaoyan 6/Long 94-4083 | 1– | 1 | 0 | – | – | – | – | – | – | – |
| Longmai 10 | Heilongjiang | Dongnong 101/Yuanzhong 3908 | 0 | 0 | 0 | – | – | – | – | – | – | – |
| Longmai 15 | Heilongjiang | Ke 76-686/Tieling 3 | 1 | 1+ | ; | – | – | – | – | – | – | – |
| Longmai 20 | Heilongjiang | Unknown | 0 | 0 | ; | – | – | – | – | – | – | – |
| Longmai 23 | Heilongjiang | Unknown | 0 | 0 | 0 | – | – | – | – | – | – | – |
| Longmai 24 | Heilongjiang | Unknown | 0 | 0 | 0 | – | – | – | – | – | – | – |
| Longmai 26 | Heilongjiang | Long 87-7129/Ke 88F22060 | ;1 | 0 | 0 | + | – | – | – | – | – | + |
| Longmai 27 | Heilongjiang | Unknown | ;1 | 1 | 0 | – | – | – | – | + | + | – |
| Longmai 30 | Heilongjiang | Long 90?05098/Long 90? 06351 | 1 | 0 | 1 | – | – | – | – | – | – | + |
| Longmai 31 | Heilongjiang | Longmai 20/PSN/BOW//Longmai 206 | 0 | 0 | 0 | – | – | – | – | – | – | – |
| Longmai 32 | Heilongjiang | Long 94-4018/Ke 88F$_2$165-3 | 0 | 0 | 0 | – | – | – | – | – | – | – |
| Longmai 33 | Heilongjiang | Longmai 26/Jiusan 3u92 | ; | ; | 0 | + | – | – | – | – | – | – |
| Longmai 34 | Heilongjiang | F$_1$ (Zhong B054-3/2*Longmai 15//97 Chanjian489/3)/Longmai 26 | ; | ; | ; | + | – | – | – | – | – | – |
| Longmai 35 | Heilongjiang | Ke 90-513/Longmai 26 | 1– | 0 | 0 | – | – | – | – | – | + | + |
| Longmai 36 | Heilongjiang | Ke 92-387/Long 99F$_3$-6725-1 | 0 | 0 | 1– | – | – | – | – | – | – | – |

(Continued)

| Cultivars | Province | Pedigree | Infection types[a] | | | Sr2 | Sr24 | Sr25 | Sr26 | Sr31 | | Sr38 |
|---|---|---|---|---|---|---|---|---|---|---|---|---|
| | | | 21C3CTHQM | 34MKGQM | 34C3RTGQM | Xgwm533 | Sr24#12 | Gb | Sr26#43 | SCSS30.2$_{576}$ | Iag95 | VENTRIUP-LN2 |
| Longmai 37 | Heilongjiang | Long 2003M8059-3/Long 01D1572-2 | ; | 1 | 0 | + | – | – | – | – | – | – |
| Longmai 39 | Heilongjiang | Long 03F3-6519/Longfu 20-378 | 2 | 2 | 0 | + | – | – | – | – | – | – |
| Kefeng 2 | Heilongjiang | Kehan 7/Ke 68F$_4$-585-13 | 0 | 1– | 1 | + | – | – | – | – | – | – |
| Kefeng 3 | Heilongjiang | Kehan 8/Kehong/Kezheng//Nadadoles | ; | 1– | ; | – | – | – | – | – | – | – |
| Kefeng 4 | Heilongjiang | Ke 71F4-370-7/Moyi 66 | 0 | 0 | ; | – | – | – | – | – | – | – |
| Kefeng 5 | Heilongjiang | Ke 76-250/Ke 76F_4-799-5 | 1 | 0 | 0 | – | – | – | – | – | – | – |
| Kefeng 6 | Heilongjiang | Ke 85-869/Ke 85-784 | ;1 | 2 | 1– | – | – | – | – | – | – | + |
| Kefeng 10 | Heilongjiang | Kehan 12/Ke 89RF$_6$287 | 0 | 0 | ; | – | – | – | – | – | – | – |
| Kenda 4 | Heilongjiang | 82-5621/Ke 79-369 | 0 | 0 | 0 | – | – | – | – | – | – | – |
| Kenda 5 | Heilongjiang | Longfu 5009/Nongda 84-838 | 0 | 0 | 0 | – | – | – | – | – | – | – |
| Kenda 6 | Heilongjiang | Nongda 89-2729/Bei 89-22 | 0 | 1– | 0 | – | – | – | – | – | – | – |
| Kenda 7 | Heilongjiang | Nongda 89-2729/Bei 89-22 | 0 | 0 | 0 | – | – | – | – | – | – | – |
| Kenda 8 | Heilongjiang | Nongda 89-2729/Bei 86-1701 | 0 | 0 | 0 | – | – | – | – | – | – | – |
| Kenda 9 | Heilongjiang | Nongda 88-1116-8/Bei88-26 | 2 | 1 | 1– | – | – | – | – | – | – | – |
| Kenda 10 | Heilongjiang | Nongda 94-3537/Bei 90-1201 | 1 | 1 | 1 | – | – | – | – | – | – | – |
| Kenda 11 | Heilongjiang | Jiusan 93u92/Ke 90-514 | 0 | 0 | 0 | – | – | – | – | – | – | – |
| Kenda 12 | Heilongjiang | Jiadongmai 19/Nongda 96-2543 | 0 | 1 | 1– | – | – | – | – | – | – | – |
| Kenda 13 | Heilongjiang | Unknown | 0 | 0 | 0 | + | – | – | – | – | – | – |
| Kenjiu 9 | Heilongjiang | Xiyin 1/Jiusan 80-41123-7-3 | 0 | 0 | 0 | – | – | – | – | + | + | – |
| Kenjiu 10 | Heilongjiang | Jiusan 84-7251/Jiusan 87148//Ke | 0 | 0 | 1– | – | – | – | – | – | – | + |
| Kechun 2 | Heilongjiang | Ke 90-514/Ke 93RF$_6$-128//Ke 90-514 | ; | 1 | 1– | + | – | – | – | – | – | + |
| Kechun 5 | Heilongjiang | Ke 99F2-33-3/Jiusan 94-9178 | 0 | 0 | 0 | – | – | – | – | + | + | + |
| Kechun 8 | Heilongjiang | Ke 99F$_2$-33-3/Jiusan 94-9178 | 0 | 0 | 0 | – | – | – | – | + | + | + |

| Cultivars | Province | Pedigree | Infection types[a] | | | Sr2 | Sr24 | Sr25 | Sr26 | Sr31 | Iag95 | Sr38 |
|---|---|---|---|---|---|---|---|---|---|---|---|---|
| | | | 21C3CTHQM | 34MKGQM | 34C3RTGQM | Xgwm533 | Sr24#12 | Gb | Sr26#43 | SCSS30.2$_{576}$ | Iag95 | VENTRIUP-LN2 |
| Kechun 9 | Heilongjiang | Ke 99F$_2$-33-3/Jiusan 94-9178 | 0 | 1+ | 2 | – | – | – | – | + | + | – |
| Xiaobing 33 | Heilongjiang | A. glaucum/Triticum aestivum | 0 | 0 | 0 | – | – | – | – | – | – | – |
| Beimai 6 | Heilongjiang | Jiusan 93-3U92/Ke 90-514 | 0 | 2 | 0 | – | – | – | – | – | – | + |
| Beimai 9 | Heilongjiang | Jiusan 97F$_4$-1057/Jiusan 97F$_4$-255F]/119-54-4--3 | 2 | 1 | 1 | – | – | – | – | – | – | + |
| Longken 402 | Heilongjiang | Unknown | 1 | 1 | 0 | – | – | – | – | – | – | – |
| 2010j159 | Heilongjiang | Unknown | 0 | 0 | 2 | + | – | – | – | + | + | – |
| Norstar | Heilongjiang | Unknown | 1 | 2 | 1+ | + | – | – | – | – | – | + |
| Dongnong 125 | Heilongjiang | Unknown | ;1 | 0 | 2 | – | – | – | – | – | – | + |
| Nongmai 850 | Beijing | Unknown | 1 | 0 | 0 | + | – | – | – | + | + | – |
| Zhongmai 8 | Beijing | Hehua 971-3/Ji Z76 | 1+ | 1 | 0 | – | – | – | – | – | – | + |
| Jingdong 8 | Beijing | Afuleer 5238-016/Hongliang 4//Jingnong 79-106 | 2 | 0 | 1 | – | – | – | – | + | + | – |
| Zhongmai 895 | Beijing | Zhoumai 16/Liken 4 | 1 | 0 | 1 | – | – | – | – | + | + | – |
| Chimai 2 | Inner Mongolia | Wenge 7/Kehan 6 | 1 | 0 | 1 | – | – | – | – | – | – | – |
| Chimai 5 | Inner Mongolia | Wenge 1/Ke 76 tiao 295 | ;1 | 1 | 1– | – | – | – | – | – | – | – |
| Chimai 7 | Inner Mongolia | Ke 76 tiao 295/Wenge 1 | 2 | 1 | 1 | – | – | – | – | – | – | – |
| Ba 13p51 | Inner Mongolia | Unknown | ;1– | 0 | 1 | + | – | – | – | + | + | – |
| Shannong 22 | Shandong | PH82-2-2/954072 | 2 | 1+ | 2 | – | – | – | – | – | – | – |
| Shannong 23 | Shandong | Tal (Ms2) recurrent selection | 2 | 2 | 2 | – | – | – | – | + | + | – |
| Shannong 24 | Shandong | Tal (Ms2) recurrent selection | 1– | 0 | 2 | – | – | – | – | + | + | – |
| Jimai 19 | Shandong | Lunai 13/Linfen 5064 | 4 | 4 | 4 | + | – | – | – | – | – | – |
| Jimai 20 | Shandong | Lunai 14/Lu 884187 | ; | 0 | 0 | + | – | – | – | – | – | + |
| Jimai 21 | Shandong | 865186/Chuanmongda 84-1109/Ji 84-5418 | 3 | 4 | 4 | – | – | – | – | – | – | – |
| Jimai 22 | Shandong | 935024/935106 | 0 | 0 | 1 | – | – | + | – | – | – | + |
| Jimai 44 | Shandong | Jinan 17/954027 | 2 | ; | 2 | + | – | – | – | + | – | + |
| Yannong 19 | Shandong | Yan 1933/Shan 82-29 | ;1 | ; | ; | – | – | – | – | – | – | + |
| Yannong 21 | Shandong | Heyan 1933/Shan 8229 | 3– | 2 | 1 | – | – | – | – | – | – | – |
| Yannong 23 | Shandong | Yan 1061/Lumai 14 | 3 | 4 | 3 | – | – | – | – | – | – | – |
| Tainong 18 | Shandong | Laizhou 137/Yan 369-7 | 2 | 0 | 1 | – | – | – | – | – | – | – |

(Continued)

| Cultivars | Province | Pedigree | Infection types[a] | | | Sr2 | Sr24 | Sr25 | Sr26 | Sr31 | | Sr38 |
| --- | --- | --- | --- | --- | --- | --- | --- | --- | --- | --- | --- | --- |
| | | | 21C3CTHQM | 34MKGQM | 34C3RTGQM | Xgwm533 | Sr24#12 | Gb | Sr26#43 | SCSS30.2$_{576}$ | Iag95 | VENTRIUP-LN2 |
| Taishan 23 | Shandong | 876161/881414 | 1+ | 0 | 1– | – | – | – | – | + | + | – |
| Taishan 24 | Shandong | 904017/Zhenzhou 8329 | 3 | 4 | 3– | – | – | – | – | – | – | – |
| Luyuan 502 | Shandong | 9940168/Jimai 19 | ; | 0 | 0 | – | – | – | – | + | + | – |
| Tanmai 98 | Shandong | Jining 13/942 | 4 | 3 | 2 | – | – | – | – | – | – | – |
| Lumai 21 | Shandong | Yanzhong 144/Baofeng 7228 | 3+ | 4 | 4 | – | – | – | – | – | – | – |
| Jinan 17 | Shandong | Linfen 5064/Lumai 13 | 4 | 4 | 4 | – | – | – | – | – | – | – |
| Liangxing 66 | Shandong | Ji91102/Ji 935031 | 2 | 3 | 1 | – | – | – | – | – | – | – |
| Liangxing 99 | Shandong | Ji 91102/Lumai14//PH85-16 | 3– | 4 | 0 | – | – | – | – | – | – | – |
| Zhoumai 28 | Henan | Zhoumai 18/Zhoumai 22//Zhou 2168 | 1 | ; | ; | – | – | – | – | + | + | – |
| Zhumai 762 | Henan | Unknown | 2 | ; | ; | – | – | – | – | + | + | + |
| Luomai 6010 | Henan | Yuanyang /Luo152/82C6/M | 2 | ;1 | 2 | – | – | – | – | – | – | – |
| Guomai 301 | Henan | G883/Pumai 9 | 2 | 0 | 2 | – | – | – | – | – | – | + |
| Zhoumai 27 | Henan | Zhoumai 16/Aikang 58 | 1 | 0 | ; | – | – | – | – | + | + | – |
| Xumai 33 | Henan | Neixiang 991/Zhoumai 16 | 0 | 1 | 0 | – | – | – | – | + | + | – |
| Xinmai 29 | Henan | Yanzhan 4110/Zhoumai 16 | 4 | 4 | 4 | – | – | – | – | – | – | – |
| Annong 0711 | Henan | Yannong 19/Aanong 0016 | 2 | 1 | 2 | – | – | – | – | – | – | – |
| Anke 157 | Henan | Taishan 241/Xinong 1718 | 2 | 3 | 4 | – | – | – | – | – | – | – |
| Pumai 053 | Henan | Bainong AK58/Zhoumai 18 | 1 | 1 | 0 | – | – | – | – | + | + | – |
| Kaimai 22 | Henan | Zhoumai 18/ Bainong AK58 | 1– | 1 | 0 | – | – | – | – | + | + | – |
| Zhenmai 1860 | Henan | Unknown | 1 | ; | ; | – | – | – | – | + | + | – |
| Womai 9 | Henan | Laizhou 953/Bainong AK58 | 1 | 1 | 1 | – | – | – | – | + | + | – |
| Ning 52 | Ningxia | Yong 403/Yongliang 15//Yong 1147/230 | 0 | 0 | 0 | + | – | – | – | + | + | – |
| Ning 39 | Ningxia | Yong 833/Ningchun 4 | ; | 0 | 0 | + | – | – | – | – | – | – |
| Ningchun 4 | Ningxia | Suonuola 64/Hongtu | 0 | 0 | 0 | – | – | – | – | – | – | – |
| Ning 51 | Ningxia | Yong 3002/Ningchun 4 | 0 | 0 | 0 | + | – | – | – | – | – | – |

| Cultivars | Province | Pedigree | Infection types[a] | | | Sr2 | Sr24 | Sr25 | Sr26 | Sr31 | | Sr38 |
|---|---|---|---|---|---|---|---|---|---|---|---|---|
| | | | 21C3CTHQM | 34MKGQM | 34C3RTGQM | Xgwm533 | Sr24#12 | Gb | Sr26#43 | SCSS30.2$_{576}$ | Iag95 | VENTRIUP-LN2 |
| Ningchun 53 | Ningxia | Ningchun 39/Moxige M7021 | 0 | 0 | 0 | + | − | − | − | − | − | − |
| Ningdong 11 | Ningxia | RENAN//Beinong 2/Beijing 841 | 1+ | 0 | 0 | − | − | − | − | + | + | + |
| Linfeng 3 | Shanxi | Linyuan 86-7065/Linyuan 81-5011 | 2 | 0 | ; | − | − | − | − | − | − | − |
| Jinmai 90 | Shanxi | Jinmai 47/02L013 | 4 | 4 | 4 | − | − | − | − | − | − | − |
| Wanmai 38 | Jiangsu | Yanzhong 114/85-15-9 | 1+ | 0 | 2 | − | − | − | − | − | − | + |
| Wansu 0217 | Jiangsu | Unknown | 2 | 2 | 1 | − | − | − | − | − | − | − |
| Huaimai 4064 | Jiangsu | Unknown | 1 | 1 | 1 | − | − | − | − | + | + | − |
| Wanmai 1643 | Jiangsu | Unknown | 0 | 0 | 0 | + | − | − | − | − | − | − |

**Notes:**

Seedling infection types produced by three races of *P. graminis* f. sp. *tritici* and molecular detection of resistance genes *Sr2*, *Sr24*, *Sr25*, *Sr26*, *Sr31*, and *Sr38* in the 136 wheat cultivars (lines).

[a] Infection types (ITs): are based on a 0–4 scale where ITs of 0, ;, 1, and 2 are indicative of a resistant (low) response and ITs of 3 or 4 of a susceptible (high) response; Symbols + and − indicate slightly larger and smaller pustule sizes, respectively (*Stakman, Stewart & Loegering, 1962*).

[b] Symbol "+" indicates the cultivar (line) carry the tested genes; "−" indicates the cultivar (line) don't carry the tested genes.

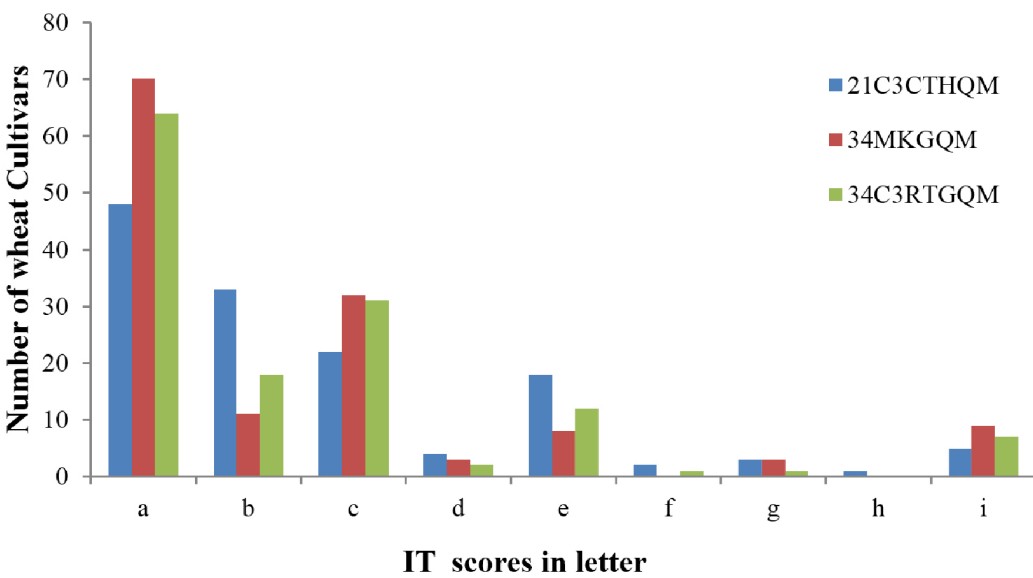

**Figure 2 Screening for resistance genes against three *Pgt* races in wheat seedlings.** Seedling infection type (IT) scores have been converted to letters to facilitate reading: a, 0; b, −1; c, 1; d, 1+; e, 2; f, 3−; g, 3; h, 3+; i, 4.

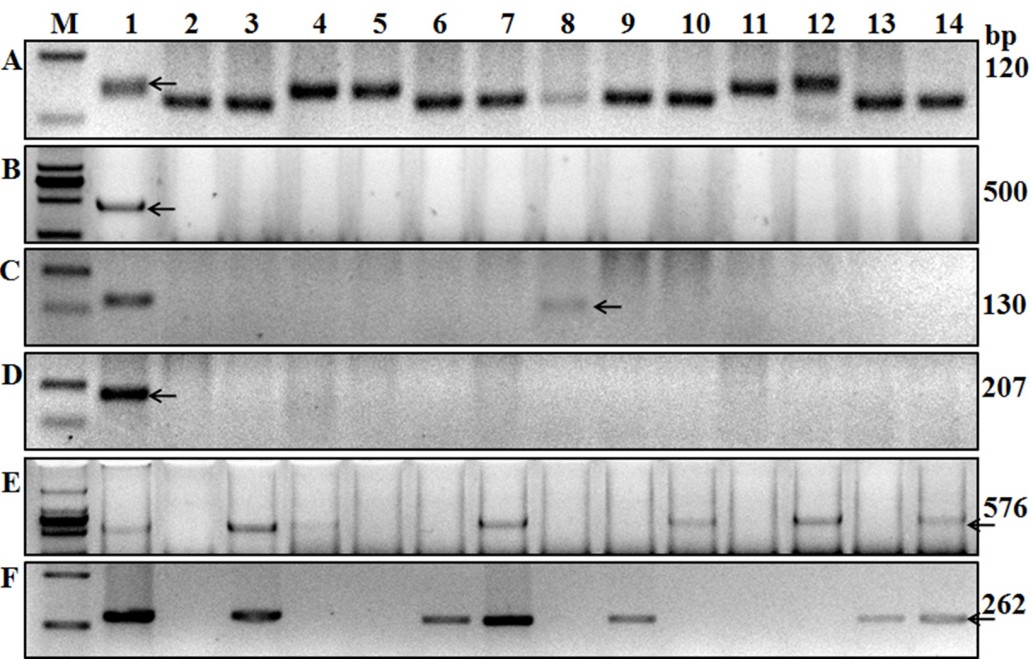

**Figure 3 Amplification results for some of the wheat cultivars tested using six markers.** A, Xgwm533; B, Sr24#12; C, Gb; D, Sr26#43; E, SCSS30.2576; F, VENTRIUP-LN2. Lanes A1 to F1 are results of Hope, LcSr24Ag, Agatha/9*LMPG, Eagle, Sr31/6*LMPG, and Trident cultivars. Lanes 2–14 are Kenda 9, Kechun 8, Nongmai 850, Longfu 18, Kenjiu 9, Ningdong 11, Jimai 22, Yannong 19, Taishan 23, Ning 39, Ning 52, Wanmai 38, and Zhumai 762 cultivars. M is the DNA ladder used to identify the specific sequences of each molecular marker.

### *Sr31* and *Sr38* screening

The effective resistance of *Sr31* and *Sr38* to *Pgt* was overcome by Ug99, as no race with virulence to these genes had been found in China (*Li et al., 2016*). Genes *Sr31* and *Sr38* were widely used in wheat programs. Markers $SCSS30.2_{576}$ (amplifies a 576-bp fragment) and *Iag95* (amplifies a 1,100-bp fragment) linked to *Sr31*, and the 2NS-specific primer *VENTRIUP-LN2* (amplifies a 262-bp fragment), linked to the rust resistance gene cluster *Lr37-Sr38-Yr17*, were used in the present study to screen *Sr31* and *Sr38*. Fragment sizes consistent with the presence of both resistant genes were amplified in 25 wheat cultivars and in the positive control Sr31/6*LMPG using markers $SCSS30.2_{576}$ and *Iag95*, and in 28 wheat cultivars using marker *VENTRIUP-LN2*.

## DISCUSSION

It is reported that the resistance of wheat cultivars to *Pgt* is higher in the northern rather than in the southern wheat region, especially in varieties from North China where stem rust is prone to occur. Results obtained in the present study are similar to that previously reported. In total, 124 (91.2%) wheat cultivars were resistant to the three *Pgt* races (ITs 2, 1+, or lower), and the resistance level of the accessions from Heilongjiang was higher than that of accessions from other provinces. All wheat cultivars from Heilongjiang Province were resistant to races 21C3CTHQM, 34MKGQM, and 34C3RTGQM, as wheat lines must be resistant to *Pgt* for being registered in Heilongjiang. In addition, the resistance level of wheat lines from Heilongjiang is tested by the Plant Immunity Institute, Shenyang Agricultural University, every year using the 21C3 and 34 *Pgt* race groups. Therefore, all registered cultivars registered in Heilongjiang should present ITs below 3, which was confirmed in the present study (0, 1, 1-, and 2 ITs were found; Table 2). Wheat cultivars from the lower-middle Yangtze River basin and central winter-wheat growing provinces were also highly resistant (73.9%) to the tested *Pgt* races.

Gene *Sr2*, originated from *Triticum dicoccum* Schronk, was transferred into North American and International Maize and Wheat Improvement Center (CIMMYT) wheat breeding programs in 1925, and since then it has been extensively used in many regions worldwide (*Borlaug, 1968*). In the present study, marker *Xgwm533*, which was used to detect *Sr2*, revealed that only 21 of the 136 wheat varieties were likely to carry this gene. Such cultivars might be resistant to Ug99, as the high resistance of Jimai 20 to Ug99 tested in Kenya in 2006 has been attributed to the *Sr2* gene carried by this cultivar (*He, Xia & Chen, 2008*). But it is difficult to conclude that these 21 wheat varieties carry *Sr2*, because many North American, Australian and CIMMYT lines which predicted not carry this gene can amplified a 120-bp fragment (*Jemanesh et al., 2013*; *Mago et al., 2011*).

The wheat stem rust gene *Sr24*, derived from *T. ponticum*, is effective against most *Pgt* races, including race TTKSK (i.e., Ug99). Races virulent to *Sr24* are rare in the *Pgt* population in North America (*Jin et al., 2008*). This gene has been used as a differential in North America and worldwide race surveys, but a new variant of race TTKSK with *Sr24* virulence has arisen in Kenya, South Africa, Tanzania, Ethiopia, Mozambique, and Uganda (*BGRI, 2017*). Leaf rust gene *Lr24* in association with *Sr24* provides resistance to all *Pgt* isolates. Thus, we used marker *Sr24#12*, completely linked to *Sr24* (*Mago et al.,*

2005), to screen for *Sr24*/*Lr24* genes in the 136 wheat accessions. None of the tested cultivars carried *Sr24*, although previous research using postulated gene presence based on marker data showed that some Chinese wheat cultivars might be carrying this gene (*Cao et al., 2007*). On the other hand, the study conducted by *Zhang et al. (2008)* supports our result as none of the 23 wheat cultivars they screened using molecular markers linked to *Lr24* carried this gene. Thus, more races and molecular markers should be used to confirm whether Chinese wheat cultivars carry *Sr24* or not.

Genes *Sr25* and *Sr26,* derived from *T. ponticum*, are effective against Ug99 and all *Pgt* races in China, and their use is increasing based on their resistance to Ug99 (*Bariana et al., 2007*). Novel genetic tools based on molecular marker technologies were developed to tag the presence of those genes (*Mago et al., 2005*; *Liu et al., 2010*). In the present study, we used molecular markers *Gb*, linked with *Sr25*, and *Sr26#43*, linked with *Sr26*, to identify these genes. Two wheat cultivars, "Kehan 3" (Ke61F$_3$-199/*Agropyronglaucum*) and "Jimai 22" (935024/935106) are likely to carry *Sr25*. Pedigree tracking indicated that "Kehan 3" parents contained *A. glaucum*, but *Sr25* is derived from *T. ponticum*, so the result obtained for this cultivar might not be accurate. Expectedly, none of the wheat varieties carried *Sr26*, as this gene is not widely used in breeding programs in China (*Li et al., 2016*). Results obtained here are similar to those of previous studies; for example, using marker *Sr26#43*, *Li et al. (2016)* detected that none of the 119 wheat materials examined carried *Sr26*.

Gene *Sr31*, derived from "Petkus" rye, is located on 1BL/1RS. It is distributed in wheat cultivars worldwide, but was transferred into Chinese wheat backgrounds from the Soviet Union and Romania in the 1960s (*Jiang et al., 2007*). Since then, the wheat cultivars "Alondra S," "Aftab LeEr," "Kavkaz," and "Luofulin" lines carrying *Sr31* have been released in wheat growing regions in China. Although, this gene is susceptible to Ug99, it is effective against TKTTF and TTTTF and all *Pgt* races in China (*Pretorius et al., 2000*; *Olivera et al., 2015*; *Bhattacharya, 2017*; *Li et al., 2016*). Markers *Iag95* and *SCSS30.2$_{576}$*, which were used to screen the gene *Sr31* in the present study, revealed that 25 wheat cultivars contained *Sr31*, and pedigree information and low ITs supported these results. Thus, *Sr31* should be used in breeding programs in China in combination with other genes resistant to Ug99 to ensure that Chinese wheat cultivars are resistant to Chinese *Pgt* races and to Ug99.

Gene *Sr38*, originated from *T. ventricosum*, is widely used due to its association with the stripe rust gene *Yr17* and the leaf rust gene *Lr37* that confer resistance to the three species of wheat rust pathogens (*Delibes et al., 1993*; *Dyck & Lukow, 1988*). Genes *Yr17* and *Lr37* were reported from wheat cultivars in China using molecular markers linked to them (*Peng et al., 2013*; *Xue et al., 2014*). In the present study, the marker *VENTRIUP-LN2*, which is linked with the *Sr38-Yr17-Lr37* cluster of rust resistance genes, was used and the specific PCR fragment for this marker was detected in 28 of the 136 wheat cultivars examined. These 28 cultivars presented low ITs indicating they carry *Sr38*. Gene *Sr38* is susceptible to Ug99, similar to gene *Sr31*, but resistant to all *Pgt* races in China (*Cao et al., 2007*). Therefore, in China, it should be used in combination with genes resistant to Ug99 through gene pyramiding.

Molecular markers linked to resistance genes are an alternative to gene postulation and may allow breeders to identify resistance genes rapidly and accurately (*Goutam et al., 2013*). Combining molecular markers with pedigree information of the tested varieties can greatly increase the success of gene postulation (*Yu et al., 2010*). Due to the rapid development of molecular markers and to the great importance of the new *Pgt* races, molecular markers closely linked to resistance genes against such races have been frequently reported, and many have been converted to simple sequence repeat (*Mago et al., 2013*; *Tsilo, Jin & Anderson, 2007*), sequence tagged site/cleaved amplified polymorphic sequence (*Helguera et al., 2003*; *Mago et al., 2011*), sequence tagged site (*Mago et al., 2005*; *Bansal et al., 2014*), and simple sequence repeats/amplified fragment length polymorphism markers (*Periyannan et al., 2014*). This approach overcomes gene interactions and plant stage-dependent gene expression problems associated with traditional gene postulation.

## CONCLUSION

In the present study, we used molecular markers to determine if *Sr2*, *Sr24*, *Sr25*, *Sr26*, *Sr31*, and *Sr38* were present in the 136 wheat cultivars examined. Overall, genes *Sr31*, and *Sr38* were differently distributed across wheat regions in China and none of the wheat cultivars contained *Sr24* and *Sr26*. Additional studies will be needed to verify the gene postulations for *Sr2* and *Sr25*. These cultivars comprising stem rust resistance genes are valuable genetic materials for future wheat-breeding plans.

## ACKNOWLEDGEMENTS

We appreciate very much Dr. Qingjie Song, Dr. Hongji Zhang, and Dr. Yantai Guo at Heilongjiang Academy of Agricultural Science; Dr. Jiandong Han at Shandong Academy of Agricultural Science, M.S. Zhaojie Luan at Zhenjiang Agricultural Committee; M.S. Lulan Shen at Tunliu Agricultural Committee; M.S. Xuetao Sun at Inner Mongolia grassland Bureau for providing the wheat cultivars.

### Funding
This work was supported by the grants from the National Natural Science Foundation (31701738) and Technology Research Project of Education Department of Liaoning (LSNYB201614). The funders had no role in study design, data collection and analysis, decision to publish, or preparation of the manuscript.

### Grant Disclosures
The following grant information was disclosed by the authors:
National Natural Science Foundation: 31701738.
Technology Research Project of Education Department of Liaoning: LSNYB201614.

### Competing Interests
The authors declare that they have no competing interests.

## Author Contributions

- Xiaofeng Xu performed the experiments, authored or reviewed drafts of the paper.
- Depeng Yuan prepared figures and/or tables.
- Dandan Li performed the experiments.
- Yue Gao performed the experiments.
- Ziyuan Wang performed the experiments.
- Yang Liu analyzed the data, prepared figures and/or tables.
- Siting Wang analyzed the data, prepared figures and/or tables.
- Yuanhu Xuan conceived and designed the experiments, contributed reagents/materials/analysis tools, authored or reviewed drafts of the paper, approved the final draft.
- Hui Zhao prepared figures and/or tables.
- Tianya Li conceived and designed the experiments, analyzed the data, contributed reagents/materials/analysis tools, authored or reviewed drafts of the paper, approved the final draft.
- Yuanhua Wu conceived and designed the experiments.

## Data Availability

The raw data are provided in a Supplemental File.

## Supplemental Information

Supplemental information for this article can be found online at http://dx.doi.org/10.7717/peerj.4882#supplemental-information.

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
