# Peer review of "Identification of stem rust resistance genes in wheat cultivars in China using molecular markers"

_PeerJ, doi:10.7717/peerj.4882_

## Round 0.1 · original submission · Minor Revisions

In general the manuscript was well prepared; however, there was some insight that focused on the findings and how they may be compared to prior works. Since the Xu et al. (2017) paper is cited extensively how is the experimental design different and what other gene sets are measured in this study; a more explicit explanation would be of value. Another concern was the issue of replication at the seedling stages which may ultimately influence the significance of the findings. As such, findings are very important to get to the public in regard to planning approaches at developing rust resistant germplasm; it would be of great value to reflect what distinguishes this work from others, and what new findings are evident, and how might this work be used for strategies in developing better germplasm. It is also important that the statements made are as accurate as possible due to the importance of the work, please follow the reviewers comments with high regard. I will return this with a suggestion of minor edits required; however, please address reviewer comments in your next revision.

Reviewer 1 ·

Basic reporting

It is written in good English.

Experimental design

Experimental design is sufficient. But as i mentioned below, please cite the previous article rather than copied some of the materials and methods part.

Validity of the findings

It is replication of Xu et al. 2017. So the authors need to state clearly the rationale and benefits of this paper in comparison to what is published.

Additional comments

Introduction
It has to be clearly stated what additional information this paper will bring to the wheat community in comparison to Xu et al. 2017 paper.


Materials and Methods
It is not clear. Are you tested a mixture of spring and winter wheat lines?
Most of the materials and methods are similar and explained in the same manner (particularly line 94-124) as indicated on Xu et al. 2017 paper. In this case, it is better to cite the paper.

Result
Table 1
Table 2


Conclusion
The article published by Xu et al. 2017 "Stem rust resistance genes evaluation and identification of Sr2, Sr24, Sr26, Sr31 2 and Sr38 in wheat lines from Gansu Province in China" has similar objective, used similar races and approach. the only difference is is the materials and the province where they are tested.

But Xu et al. 2017 concluded " The molecular markers linked to Sr2, Sr24, Sr26, Sr31, and Sr38 were used to detect the occurrence of these genes in 75 major wheat cultivars (lines) in Gansu Province in this study. The results showed that 35 tested cultivars might carry one of these genes." where as in this study it is concluded as "Overall, genes Sr31, and Sr38 were differently distributed across wheat regions in China and none of the wheat cultivars contained Sr24 and Sr26."

For me it looks a bit contradicting concerning Sr24 and Sr26. So it needs some verification.

Reference
According to PeerJ format, the references section should be sorted by author. But here it did not applied.

Annotated reviews are not available for download in order to protect the identity of reviewers who chose to remain anonymous.

·

Basic reporting

Clear and professional English is used throughout the manuscript. A revision of the manuscript is attached with several suggested English improvements. The introduction section does provide adequate background with appropriate references. Figures are mostly high-quality, though the resolution of figure 1 is a bit low. Raw data is supplied

Experimental design

The experimental design is appropriate for the conclusions and goal of the study. Methods are well described.

Validity of the findings

Though seedling screening was not replicated, the replication is not absolutely necessary for PeerJ. Some similar studies have not replicated the seedling assays. Overall, the data are robust and conclusions are largely well-supported.

Additional comments

Overall the manuscript is well done and is worth publishing. I do have three suggested revisions in addition to a revision of the manuscript with suggestions for improving English:

The sentence in lines 71-73 is not completely accurate. The previous work identified resistance genes present in select wheat lines from Chinese provinces, but new Ug99 resistance genes were not identified. Please modify the sentence accordingly.

What are the isolate names of the Pgt races used in this study? Providing this information is necessary.

In the conclusion section the authors claim that additional studies are needed to confirm Sr2 postulations (I agree with this claim). However the results section is written as if the Sr2 marker data is clearly supportive of Sr2 gene presence/absence. Please describe in the results or conclusion section why the marker data available is not sufficient to accurately determine Sr2 presence/absence.

---

## Round 0.2 · accepted · Accept

Thank you for your responses in regards to the reviewer comments. The manuscript reads well and should be ready for publication. There was really only one line (line 192) that may need some revision (see below). I think some of the background information helped provide the context for the work presented in this study. With the importance of getting ahead of the evolving stem rust populations this work helps to build strategies in developing better germplasm. Congratulations on this work, please consider it accepted.

line 193:
None of the tested cultivars carried Sr24, although previous research using gene postulation based on marker data showed that some Chinese wheat cultivars might be carrying this gene (Cao et al., 2007).

Suggested change:
None of the tested cultivars carried Sr24, although previous research using postulated gene presence based on marker data showed that some Chinese wheat cultivars might be carrying this gene (Cao et al., 2007).

#